# Paving the Path for Immune Enhancing Nutrition in Colon Cancer: Modulation of Tumor Microenvironment and Optimization of Outcomes and Costs

**DOI:** 10.3390/cancers15020437

**Published:** 2023-01-10

**Authors:** Maria Raffaella Ambrosio, Luigi Spagnoli, Bruno Perotti, Federica Petrelli, Saverio Caini, Calogero Saieva, Sofia Usai, Matteo Bianchini, Andrea Cavazzana, Marco Arganini, Andrea Amorosi

**Affiliations:** 1Pathology Unit, Azienda USL Toscana Nord Ovest, 56121 Pisa, Italy; 2Surgery Unit, Ospedale Unico Versilia, Azienda USL Toscana Nord Ovest, 56121 Pisa, Italy; 3Risk Factors and Lifestyle Epidemiology Unit, Institute for Cancer Research and Clinical Network (ISPRO), 50134 Florence, Italy; 4Department of Surgical Science, Sapienza University of Rome, 00100 Rome, Italy; 5Department of Health Sciences, “Magna Graecia” University of Catanzaro, 88100 Catanzaro, Italy

**Keywords:** immunonutrition, colorectal cancer, tumor microenvironment, macrophages, T-lymphocytes

## Abstract

**Simple Summary:**

Proper management of patients undergoing surgery for colorectal cancer is open to research. Three factors influence the immune response against cancer: the genetic makeup of the host, the somatic profile of cancer cells, and the surrounding environment. Immunonutrition has been suggested to impact tumor microenvironment and improve clinical outcomes in operated patients; however, its effects in vivo are far from being fully elucidated. Comparing immune function related indicators in tumor microenvironment before and after immunonutrient supplementation, we showed a favorable modulation towards an enhanced immune response against cancer. In addition to confirming improved short-term outcomes, we found that overall survival was prolonged in patients receiving immunonutrition. Finally, immunonutrition was proven to be a cost saving intervention. This study supports the rationale for implementing immunonutrition in the routine management of patients undergoing surgery for colorectal cancer.

**Abstract:**

Introduction. Published evidence suggests that immunonutrition has the potential to decrease postoperative complications and reduce length of stay in patients undergoing surgery for colorectal cancer. However, only a few studies have analyzed the effects of immunonutrition on tumor microenvironment and evaluated its prognostic impact. Material and methods. This is a single center retrospective study enrolling 50 patients undergoing elective surgery for colorectal cancer managed with immunonutrition and 50 patients managed with standard nutrition for comparison. Tumor microenvironment was analyzed before (on the biopsy at the time of diagnosis) and after (on the matched surgical specimen) administration of immunonutrition. Immune function related indicators, including cytotoxic T-lymphocytes, helper T-cells, antigen presenting cells, natural killer cells, T-exhausted lymphocytes, T-regulatory cells, M1 and M2 tumor associated macrophages and PD-L1 expression were assessed by immunohistochemistry. For both groups, clinicopathological data were collected and a 5-year follow-up was available. Results. We found that immunonutrition significantly activated the T-cell response against cancer, alter tumor microenvironment phenotype towards M2 polarization and inhibits the PD1/PD-L1 axis. A lower rate of postoperative complications and a shorter length of stay (*p* = 0.04) were observed in the immune nutrition group. Compared to standard nutrition group, patients managed wit immune nutrition showed a higher 5-year overall survival (*p* = 0.001). Finally, immune nutrition allowed to reduce the hospital care costs. Conclusions. Immunonutrition modulates tumor microenvironment by improving immune function and could prolong survival in patients undergoing elective surgery for colorectal cancer. Further studies are needed to optimize IN protocols and confirm their prognostic impact.

## 1. Introduction

Colorectal cancer (CRC) is the third most common cancer type worldwide, with more than 1.9 million new patients has estimated in 2020, and the second leading cause of cancer-mortality with nearly 1 million deaths [1,2]. Moreover, the International Agency of Research on Cancer (IARC) has predicted that the global burden of CRC will rise by 56% between 2020 and 2040, with more than 3 million new cases annually [3]. The estimated increase in mortality was even higher, with approximately 1.6 million deaths in 2040 worldwide [2]. Removal of non-metastatic or resectable CRC remains the mainstay of cure, as assessed by the National Comprehensive Cancer Network (NCCN) [4,5] and the European Society for Medical Oncology (ESMO) guidelines [6,7].

Although clinical outcomes in CRC have improved dramatically over the past 50 years, several factors may influence the result [8]. First, the high cellular turnover during tumorigenesis leads to a dysregulation of the host immune response. Secondly, patients are more likely to suffer from malnutrition due to the long-term tumor consumption before surgery, and insufficient nutritional intake because of anorexia, diarrhea, and intestinal obstruction [9]. Prevalence of malnutrition in CRC patients ranges from 45% to 60% and is significantly higher following radical surgery [10]. Together with the surgery-related impaired immune response, which induces a systemic inflammatory response syndrome, malnutrition accounts for most postoperative complications [8,11]. Complications, in turn, significantly worsen the short-term clinical results, length of stay (LOS) and hospital costs, impairing long-term oncological outcome and patient’s quality of life [12,13,14,15,16,17].

In the past, many efforts have been spent to prevent and reduce postoperative complications, mostly focusing on pathogen eradication and improvement in the hospital health care. More recently, interest has shifted towards strategies aimed to enhance the host defense mechanisms and modulate the inflammatory response to stress. Since ade-quate nutrition is closely associated with preserved immune function and reduced com-plications, nutritional support with administration of immunonutrients has been indi-cated as an effective approach for improving postoperative results [18].

To date, there is no universally accepted definition of immune nutrition (IN). In general terms, it means the addition of biologically active nutrients in higher doses than in conventional nutrition [18]. Formulations currently used include various combinations of arginine (Arg), omega-3 fatty acids (ω-3-FA), nucleotides, glutamine (Glu) and antioxidants [19,20,21].

Several reviews and meta-analysis have demonstrated the beneficial effects of IN by pooling results of randomized control trials [22,23,24]. IN has been shown to have the potential to reduce postoperative complications and LOS, as well as improve humoral and T-cell immune function [8,25]. To date, IN support is recommended by the European Society for Clinical Nutrition and Metabolism (ESPEN) for malnourished patients and is also included in ERAS (Enhanced Recovery After Surgery) protocols [26,27].

However, these guidelines, based on randomized controlled trials, have been criticized due to potential conflicts of interest, small patient samples and poor follow-up data [28,29,30,31,32]. Other studies failed to demonstrate that IN significantly affect survival, post-operative complications, and cost-benefit ratio [28,29,30,31,32].

These conflicting results are difficult to match due to the heterogeneity of study design, study population, patient sample, and nutritional program [28,29,30,31,32]. Therefore, indication (malnourished vs. well-nourished patients), timing, and optimal duration of IN management in patients undergoing surgery for CRC are still not fully clarified [33,34,35,36].

Of note, there are some concerns about the potential impact of IN on tumor biology and tumor spread, nor has clinical evidence been provided regarding effects on tumor microenvironment (TME) in vivo and long-term outcomes [37].

To address these uncertainties, we planned a retrospective study in patients who received IN support prior to elective surgery for CRC. The aim of the study was to investigate the impact of IN on TME by assessing cellular indicators of immune function before (on diagnostic biopsy) and after (on matched surgical specimen) the administration of immunonutrients. A case series of patients managed with conventional nutritional support was similarly evaluated for comparison. Five-year survival was assessed in both groups. Finally, a cost-benefit analysis was performed.

## 2. Materials and Methods

### 2.1. Design of the Study

This is a retrospective single center study conducted at the Surgery and Pathology Units of Azienda Sanitaria Toscana Nord Ovest (ATNO). 

Starting from January 2017 all patients who are admitted to the ATNO for colorectal surgery are routinely managed by ERAS protocol to reduce surgical stress, enhance recovery, and improve clinical. Moreover, since July 2017, this program has been implemented by multidisciplinary evaluation and nutritional consulting for IN compliance. The generally assessed parameters are unintentional weight loss, current body weight, body mass index (BMI), nutritional risk score (NRS), presence of malnutrition (according to the Global Criteria for Malnutrition—GLIM) [38].

Fifty consecutive patients who underwent elective surgery for CRC at the ATNO Surgery Unit between 1 July 2017 and 31 August 2017, managed with the IN protocol, as indicated above, were included in the study. The IN patients were compared with as many consecutive patients who received standard nutrition, collected in the two months (1 May 2017 and 30 June 2017) prior to the implementation of the IN protocol (control group).

For both groups, the exclusion criteria were emergency/urgent surgery for stenosis, bleeding or perforation, active infections, diabetes mellitus, kidney failure requiring dialysis, cirrhosis, inability to maintain oral feeding and required normal number of calories and protein intake with diet.

All patients were operated on by two colorectal surgeons.

All the comorbidities according to American Society of Anesthesiologists (ASA) score [39] as well as intra-, peri- and post-operatory complications were recorded.

A 5-year follow-up was available for all patients.

### 2.2. Nutrition

Patients in the IN group received oral supplementary formula (494 kcal/day) containing Arg, ω-3-FA and nucleotides for 14 days before surgery and 2 days after, in addition to the usual diet. Control patients consumed their regular daily diet without any other nutritional support.

### 2.3. Endoscopy, Surgery, Histology and Immunohistochemistry Studies

During endoscopy, an extended sampling of each suspected lesions is routinely performed in our Institution to provide enough diagnostic material for immunohistochemistry and molecular biology studies. At least six biopsies are taken (one in the proximal and distal margin, two in the center of the suspected lesion, and two in the in the intermediate areas) to make the biopsy sample highly representative of the entire lesion.

The surgical resection aims at complete excision of the tumor with free margins and extensive regional lymphadenectomy. The operation is started with a laparoscopic approach and, if necessary, converted to laparotomy.

For each case, histologic slides were retrieved from the archives of Pathology Unit of ATNO and reclassified by a pathologist (AMR) trained in gastrointestinal pathology according to the 2019 WHO criteria [1]. Staging was according to the 2018 AJCC TNM manual [1].

Immunohistochemistry assays were performed on representative formalin-fixed and paraffin-embedded (FFPE) sections by means of an automated staining platform, using commercially available products and according to the manufacturer’s instructions, as previously described [40]. 

The following cellular indicators of immune response were assessed: cytotoxic T-lymphocytes (CTL-expressing CD8), helper T-cells (TH-expressing CD4), antigen pre-senting cells (APC-expressing CD21), natural killer cells (NK-expressing CD56), T-exhausted (T-exh) lymphocytes (by coexpression of CD8 and PD-1), T-regulatory (T-reg) cells (by coexpression of CD4 and CD25), M1 and M2 tumor associated macrophages (TAM) (by calculating the CD163/CD68 ratio).

Immunoreactive cells were blindly scored on both biopsies and surgical specimens by two trained pathologists (MRA, SL) also using digital tools, as previously described [37]. Scores in the biopsy and matched surgical specimen were compared.

### 2.4. Cost-Benefit Analysis

Economic costs data were obtained from the ATNO cost center and compared in the two groups. Then the difference was computed.

### 2.5. Statistical Analysis

Statistical analysis was carried out by using commercially available statistical soft-ware (SPSS 24.0 for Windows SPSS Inc., Chicago, IL, USA) to calculate the association of epidemiologic and clinico-pathological characteristics between the two groups. Chi-squared and Fisher’s exact test were performed for descriptive statistics. The Mann–Whitney-U test was used to compare continuous variables not normally distributed. All *p* values were two-sided with *p* < 0.05 considered statistically significant [37].

## 3. Results

### 3.1. Demographic Data

A total of 100 patients were analyzed in this study, 50 in the IN group and 50 in non-IN group. The demographic and clinico-pathologic features are listed in Table 1. No significant differences were found between the two groups in age, gender, ASA score, number of comorbidities and cancer site, stage, histology, and mismatch repair protein (MMR) proficiency.

Higher body weight and BMI at the time of surgery were recorded in patients receiving IN (29.2 ± 6 in the IN group vs. 26 ± 4.1 in the control one). The difference was statistically significant (*p* = 0.03). However, no patients were classified as malnourished in both groups according to the GLIM criteria.

### 3.2. Effect of Immunonutrition on Clinical Outcomes

Clinical outcomes are showed in Table 2. A laparoscopic approach was performed more frequently (*n* = 48, 96%) in IN group than in non-IN group (*n* = 38, 76%) (*p* = 0.003) with a conversion rate of 4% (*n* = 2) and 24% (*n* = 12), respectively.

No statistically significant differences between two groups were found in overall complications (*p* = 0.43), infective complications (*p* = 0.66) and anastomotic leakage (*p* = 0.07).

A better trend was observed in IN patients, with a lower overall complication rate, no infectious complications, and anastomotic leaks in only 2 patients. Patients in the IN group had a lower mean LOS (6 days) compared with those in the non-IN group (8 days) (*p* = 0.04). 

In both groups, LOS was independent of the surgical procedure (laparoscopy vs. laparotomy) (*p* = 0.112). Nonetheless, patients undergoing laraposcopy seem to have a shorter LOS in spite of IN (Figure 1 and Figure 2).

Readmission was more frequent in non-IN (5 out of 50, 10%) than in IN patients (2 out of 50, 4%) (*p* = 0.03). The 180 days mortality rate was statistically different in the two groups (4% in IN group vs. 18% in non-IN group) (*p* = 0.025). One patient (2%) in the IN group and 6 patients (12%) in the control non-IN group recurred with metastasis to liver (*p* = 0.03). The overall survival rate was 1124.3 + 658.5 days in the IN group and 684.5 + 381.75 days in non-IN group (*p* = 0.001).

### 3.3. Tumor Microenvironment

Significant changes in TME were found in patients managed with immunonutrient supplementation comparing infiltrating immune cell populations before (on the biopsy) and after (on the matched surgical specimen) IN.

Immunohistochemistry revealed an increased number of CTL, TH lymphocytes, APC and NK cells (Figure 3A–D), a decrease of cells showing a T-exh and a T-reg phenotype, and M1 polarization (Figure 4A–C). Furthermore, both leucocytes and cancer cells showed a lower expression of PD-L1 (Figure 4D and Figure 5A–F).

The differences were statistically significant for all cells evaluated (*p* < 0.05).

In contrast, in patients managed with standard nutrition before surgery no significant changes in infiltrating leukocytes (*p* = 0.5) and PD-L1 expression (*p* = 0.3) were noted comparing the biopsy and the matched surgical specimen (Figure 6A–F).

Finally, when comparing surgical samples from patients in the IN group and the non-IN group, higher numbers of CTL and TH lymphocytes, NK cells, APCs, and from M2 to M1-TAM along with lower PD-L1 expression in TME were indicative of an enhanced immune response in the former.

### 3.4. Cost-Effectiveness Analysis

In the ATNO, the average refund of the hospital stay for colorectal surgery is €7017.7 per patient. In our series, including both IN and non-IN patients, the average hospital stay was 7 days, corresponding to an average cost of €1002.53 per patient. Considering that the mean LOS in the IN group was 6 and in the non-IN group it was 8, in the latter there was an additional cost of €2005.06 per patient. Five patients in the non-IN group and two in the IN group were readmitted, with a mean LOS of 5 days and a total additional cost for the entire series of €35,088.55. ERAS patients were recommended to take two bricks per day for 7 days before surgery and 2 days after. Since each brick costs €7, the total cost of IN management was €126 per patient. Overall, comparing the two groups, the savings produced by the treatment of IN was €117,011.

## 4. Discussion

Impairment of nutritional status is common in CRC and malnutrition impacts negatively on the host immune function and tissue healing, resulting in higher rates of post-operative complications and longer hospital stay [40,41,42,43]. 

Surgery also induces a temporary immune dysfunction, contributing to worsening the postoperative outcome [8,11].

Moreover, CRC patients are often elderly and frequently have comorbidities [44], further compounding the issue.

Despite some criticisms on available studies supporting the effectiveness of the IN support to enhance immune function and improve clinical outcomes after CRC surgery, IN is currently recommended in the management of CRC patients undergoing surgical treatment.

There is debate about the mechanisms by which IN improves immune response and insufficient data are available on long-term clinical outcomes. Immune function related indicators have been mainly evaluated on blood samples, whereas only a few studies have explored the effects on immune response in the microenvironment of gastrointestinal tissues [30,31,32,37].

The present study is a single center retrospective study aimed at assessing the effects of IN on TME and 5-year survival in patients undergoing elective surgery for CRC. As controls, we evaluated a series of CRC patients with similar demographic and clinico-pathologic characteristics, managed with standard nutrition.

Patients managed with IN showed shorter LOS, and lower rate of readmission and 180-days mortality, in agreement with previous studies. We did not find any differences in terms of peri-operative and post-operative complications (including infections and anastomotic leakage). Nonetheless, all the 100 patients enrolled followed the ERAS protocol and were thus granted from pre-rehabilitation. Overall survival was prolonged in IN patients and metastases were more frequent in non-IN group (*p* < 0.05).

Comparing the immune function related cell indicators before (on the biopsy at the time of diagnosis) and after (on the surgical specimen) immunonutrient supplementation, we observed an increased content of CTLs, TH cells, APCs, and NK cells, reduced T-exh and T-reg phenotypes, and a M1 polarization of TAMs. Addressing check-point inhibitor pathways, we found an enhanced expression of PD-L1, suggesting an inhibitory effect on PD-1/PD-L1 axis [45,46,47,48,49,50,51].

Conversely, in the non-IN group, no significant differences were noted in immune cell number and phenotypes before and after nutritional support. Moreover, compared with IN group, immune related indicators in TME were lower after administration of nutrients.

Overall, our results indicate that IN might modulate the TMA by enhancing immune response and reverting, at least in part, immune escape mechanisms engaged by the tumor [47,48,49,50,51,52].

Finally, our study is the first to report that patients undergoing elective surgery for CRC managed with IN have a longer 5-year survival than patients managed with standard nutrition. Although further studies are needed to confirm this finding and possibly elucidate the mechanisms involved, it is conceivable that the beneficial effects of IN on TME that we noted may have contributed significantly. CTLs and TH cells are important effectors of anti-tumor immunity [45,46] whereas the T-reg and T-exh phenotypes sustain the survival of cancer cells by inducing an anergic and exhausted anti-neoplastic immune response. M2-TAMs exert procarcinogenic effect, favoring tumor dissemination. Tumors can escape immune surveillance through immunosuppressive TME, restricting antitumor immunity. Inhibition of PD-1/PD-L1 pathway actively con-tributes to restore, at least in part, the immune response against CRC cells [49,50,51,52].

Timing of IN is of foremost importance in affecting the impact of post-operative in-flammation and immunosuppression as therapeutic levels of the nutrients must be reached in tissues pre-operatively. In our study, we confirmed previous evidence that a minimum of 7 days before and 2 days after surgery would be optimal for the intended benefit of IN on outcomes [11].

In Western countries, healthcare costs are rapidly increasing and the rising gap between costs and available resources has become a serious issue, dramatically emerged during the SARS-CoV-2 pandemic for the economic downturn [53]. Development and implementation of effective cost containment measures represent an ongoing major challenge involving all players in the healthcare system [54]. Preoperative assessment of surgical risk and actions aimed at reducing postoperative morbidity are mandatory to have a positive impact on hospital costs [54]. We showed that IN is an effective and cost-effective intervention, and that IN costs are offset by savings due to decreased postoperative complications, LOS, and readmissions. Regardless of nutritional status, the immunomodulatory nutritional intervention adopted according to current recommendations and managed within an ERAS protocol, represented a resource to support patient recovery after CRC surgery, even in patients well fed with an average net saving of €117,011.

Present study has some strengths and several limits. It is a single center retrospective investigation evaluating a small sample of patients with limited stratification. The positive effects of IN on survival should be confirmed prospectively in more numerous patient samples with extended follow up. However, a prospective case-control study is difficult to plan due to ethical reasons, as it would prevent access to proven beneficial treatments.

## 5. Conclusions

Present study provided novel insight into the effects of IN in CRC patient undergoing elective surgery for CRC. IN has been shown to enhance the immune response against cancer by positively modulating TME, decrease the rate of postoperative complications, reduce LOS, and prolong overall survival. These results could have important implications in improving clinical management and containing healthcare costs in CRC surgery. However, to draw definitive conclusions a higher number of cases with a more detailed survival analysis and possibly longer follow-up should be collected. Further studies are needed to better clarify the effects of IN on immune function and CRC patients’ survival and quality of life. In particular, we believe that the results obtained by immunohistochemistry should be integrated with an analysis of the cytokines involved in the immune reaction, such as IL-10, IL-17, IL-6, IFN-γ, TNFα as well as with the evaluation of the same lymphoid populations by citofluorimetry in the patients’ blood.

## Figures and Tables

**Figure 1 cancers-15-00437-f001:**
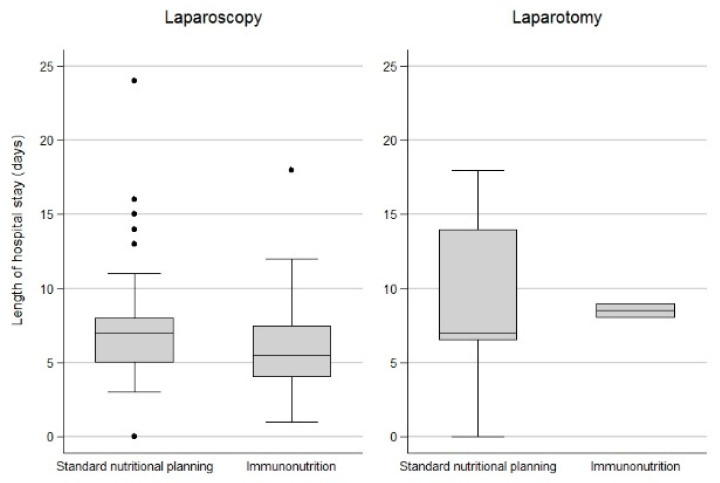
Length of hospital stay in IN and non-IN group according to surgical procedure (laparoscopy vs. laparotomy), box plot.

**Figure 2 cancers-15-00437-f002:**
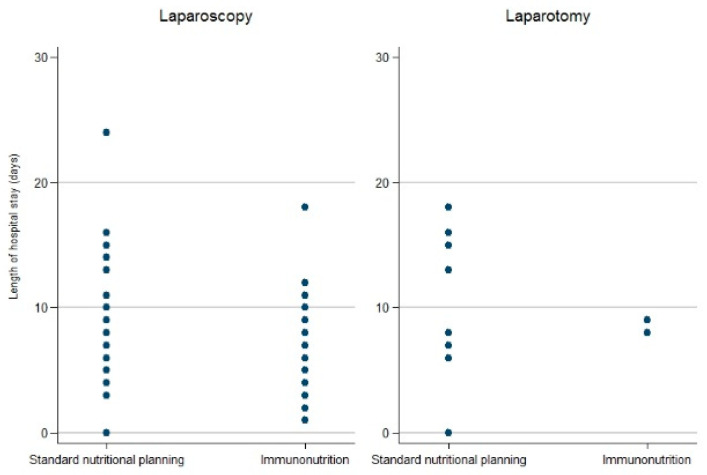
Length of hospital stay in IN and non-IN group according to surgical procedure (laparoscopy vs. laparotomy), scatter plot.

**Figure 3 cancers-15-00437-f003:**
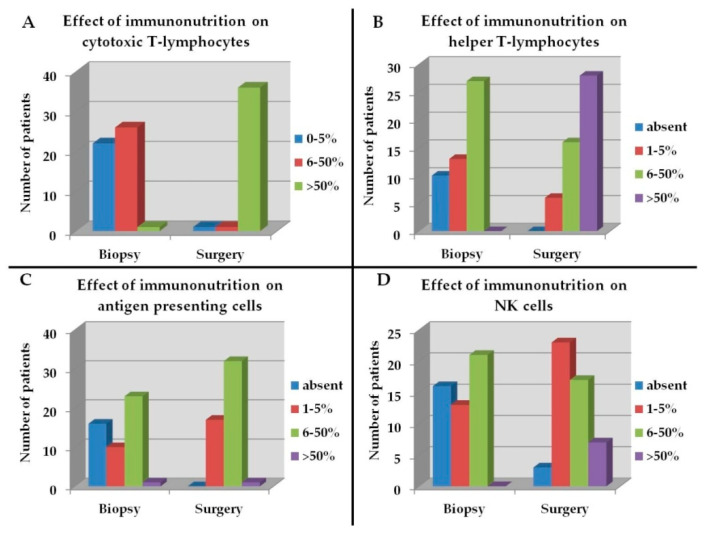
Modulation of lymphoid cells in the TME of patients receiving IN from biopsy to surgical specimen. (**A**), CTL; (**B**), TH; (**C**), APC; (**D**), NK.

**Figure 4 cancers-15-00437-f004:**
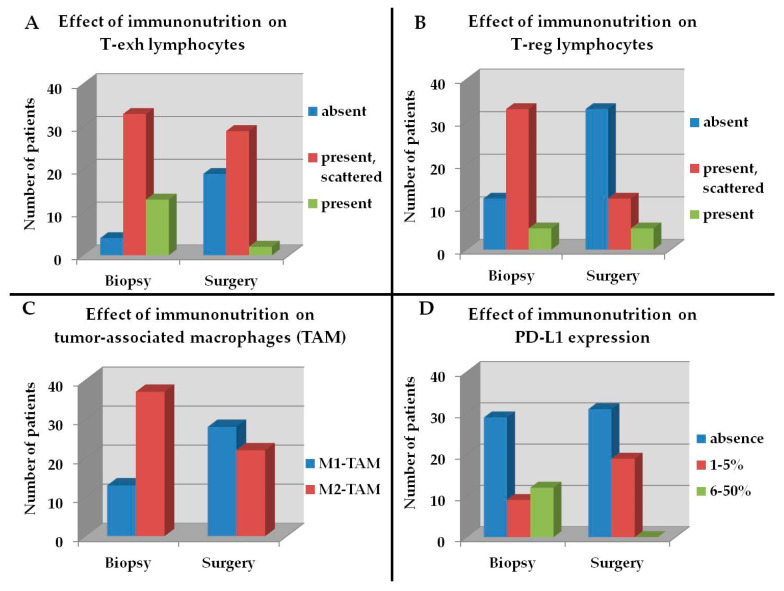
Modulation of TME of patients receiving IN from biopsy to surgical specimen. (**A**), T-exh lymphocytes; (**B**), T-reg cells; (**C**), TAM; (**D**), PD-L1 expression.

**Figure 5 cancers-15-00437-f005:**
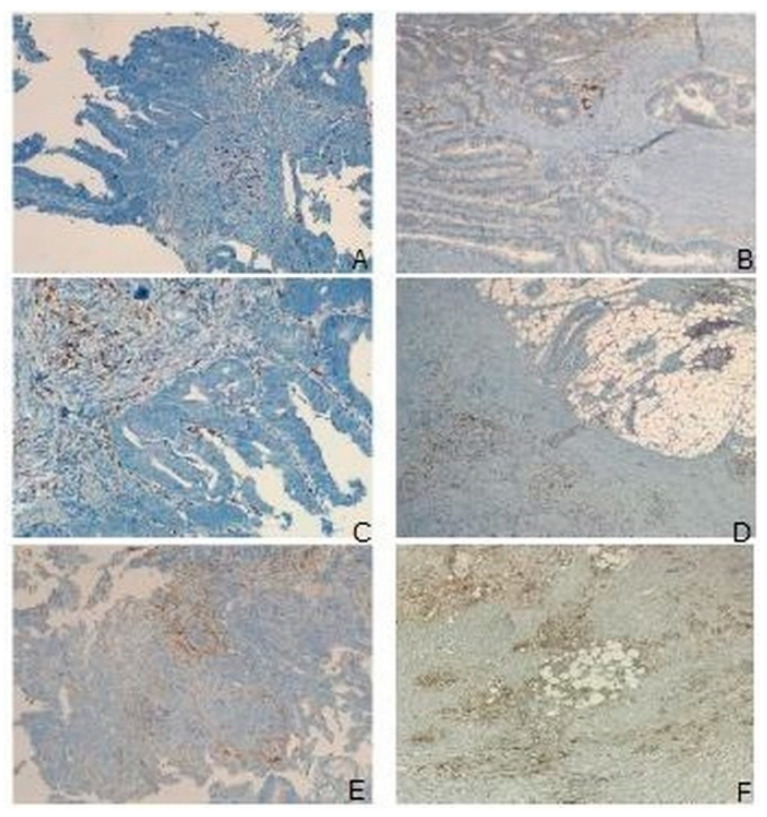
Modulation of TME in IN patients. Lower levels of infiltrating CTL and TH in the biopsy (**A**) than in the surgical sample (**B**); M2 macrophages (**C**) reverted to M1 in surgical sample (**D**). Higher expression of PD-L1 (**E**) in biopsy than in operatory specimen (**F**). (**A**–**F**), immunohistochemistry: (**A**,**B**): CD4/CD8 double stain; (**C**,**D**): CD68; (**E**,**F**): PD-L1; Original magnification (O.M.): (**A**,**C**–**F**): 10×; (**B**): 4×.

**Figure 6 cancers-15-00437-f006:**
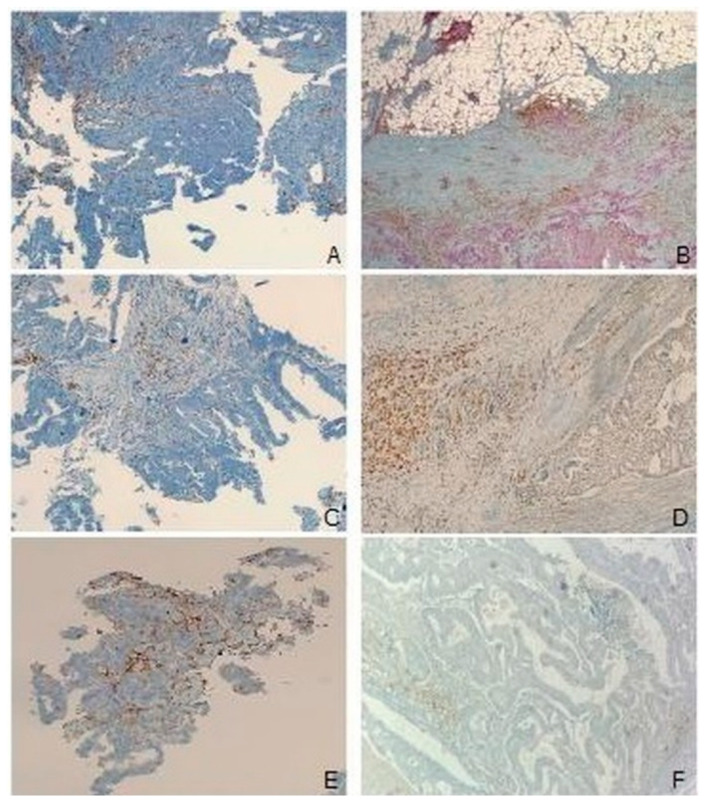
Absence of TME modulation in non-IN patients. No changes in the lymphoid population between the biopsy (**A**) and the surgical specimen (**B**); preservation of TAM polarization in the biopsy (**C**) and in the surgical specimen (**D**). Higher PD-L1 expression in both neoplastic cells and TME in the biopsy (**E**) and in the surgical specimen (**F**). (**A**–**F**), immunohistochemistry: (**A**,**B**): CD4/CD8 double stain; (**C**,**D**): CD68; (**E**,**F**): PD-L1; O.M.: (**A**,**C**–**F**): 10×; (**B**): 4×.

**Table 1 cancers-15-00437-t001:** Main features of the population under study.

Clinico-Pathologic Features	No Immunonutrition (*n* = 50)	Immunonutrition (*n* = 50)	*p* Value
**Mean age and range** (years)	63 (33–89)	69 (45–93)	0.7231
**Gender** (*n*., %)	M 24 (48%) F 26 (52%)	M 23 (47%) F 27 (53%)	0.9587
**ASA** (*n*., %) **II** **III** **IV**	13 (26%) 35 (70%) 2 (4%)	16 (28%) 33 (70%) 1 (2%)	0.6598 0.8305 0.9235
**Comorbidities** (*n*, %) **0–2** **3–4** **more than 4**	27 (54%) 17 (34%) 6 (12%)	33 (66%) 14 (28%) 3 (6%)	0.7923 0.3734 0.2687
**Site** (*n*., %) **right colon** **left colon** **rectum** **transverse colon**	29 (58%) 11 (22%) 8 (16%) 2 (4%)	22 (44%) 14 (28%) 14 (28%) 0 (0%)	0.2681 0.4884 0.1475 0.4945
**Histology**	ADK, NOS: 42 (84%) MC: 7 (14%) UC: 1 (2%)	ADK, NOS: 40 (80%) MC: 6 (12%) UC: 4 (8%)	0.3567 0.5246 0.8235
**Stage**	I: 10 (20%) IIA-B: 24 (48%) IIIA-C: 10 (20%) IVA-B: 6 (12%)	I: 12 (24%) IIA-B: 27 (54%) IIIA-C: 10 (20%) IVA-B: 1 (2%)	0.8780 0.9350 1 0.06
**MMR status**	MMR proficient: 43 (86%) MMR deficient: 7 (14%)	MMR proficient: 39 (78%) MMR deficient: 11 (22%)	0.7858 0.6571

M: male; F: female; ASA: American Society of Anesthesiologists; *n*.: number; ADK: adenocarcinoma; NOS: not otherwise specified; MC: mucinous carcinoma; UC: undifferentiated carcinoma; MMR: mismatch repair.

**Table 2 cancers-15-00437-t002:** Intra-operative and post-operative outcomes.

Clinical Outcomes	No Immunonutrition (*n* = 50)	Immunonutrition (*n* = 50)	*p* Value
**Complications according to Clavien-Dindo classification** (*n.*, %) ✓ Clavien-Dindo 0 ✓ Clavien-Dindo 1 ✓ Clavien-Dindo 2 ✓ Clavien-Dindo 3a ✓ Clavien-Dindo 3b ✓ Clavien-Dindo 4 ✓ Clavien-Dindo 5	35 (70%) 5 (10%) 5 (10%) 3 (4%) 2 (4%) 0 (0%) 0 (0%)	40 (80%) 5 (10%) 2 (4%) 2 (4%) 0 (4%) 0 (0%) 1 (2%)	0.43
**Infective complications** (*n*., %)	2 (4%)	0 (0%)	0.66
**Anastomotic leakage** (*n*., %)	3 (6%)	2 (4%)	0.07
**LOS** (days, mean)	8	6	0.04
**Readmission** (*n*., %)	5 (10%)	2 (4%)	0.03
**180 days mortality** (*n*., %)	9 (18%)	2 (4%)	0.025
**Recurrence** (*n*., %)	6 (12%)	1 (2%)	0.003
**Overall Survival** (days)	684.5 ± 381.75	1124.3 ± 658.5	0.001

LOS: length of stay.

## Data Availability

Data is contained within the article.

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
