# Peer review of "Paving the Path for Immune Enhancing Nutrition in Colon Cancer: Modulation of Tumor Microenvironment and Optimization of Outcomes and Costs"

_cancers, 2023, doi:10.3390/cancers15020437_

Round 1

Reviewer 1 Report

The presented results are not clear – please add some more details like dot charts  for all figures

-        Please add separate analysis of LOS for patients in non  IN group and laparoscopic approach and non-laparoscopic approach vs IN group  – are there any chances that non-laparoscopic approach could affect the results ? Please add dot chart (laparoscopic and non-laparoscopic approach should be marked separately).

-     Please  clarify:

Ln 228: The mean LOS in IN and non IN group was 7.

Ln 229:  ..the mean LOS in IN cohort was 6 and in non IN  cohort was 9..

Author Response

Dear Review thank you very much for your criticisms that have improved the manuscript. We aimed to address all your issues and better described material and methods and result sections as you suggested. Moreover English language was reviewed by a native-speaker.

- The presented results are not clear – please add some more details like dot charts  for all figures

We added bot plot, scatter plot and histograms to better show the effect of immunonutrition on clinical outcomes  and TME

  •        Please add separate analysis of LOS for patients in non  IN group and laparoscopic approach and non-laparoscopic approach vs IN group  – are there any chances that non-laparoscopic approach could affect the results ? Please add dot chart (laparoscopic and non-laparoscopic approach should be marked separately).

We add figures 1 and 2. By stratyfying both groups for the surgical approach we found thet LOS was independent of the surgical procedure (laparoscopy versus laparotomy) (p=0.112). Nonetheless, patients undergoing laraposcopy seem to have a shorter LOS in spite of IN (Figure 1-2).

-     Please  clarify:

Ln 228: The mean LOS in IN and non IN group was 7.

Ln 229:  ..the mean LOS in IN cohort was 6 and in non IN  cohort was 9.

We did it.

Reviewer 2 Report

In this very interesting study, the association between immune-enhancing nutrition and several outcomes in colorectal cancer patients, including the immune response, post-operative complications and survival was investigated. This is a intriguing and important topic. Although the sample size was rather small (50 cases and 50 controls), the study provide important insights into the relationship between immuno-nutrition and CRC outcomes. An important strength of this study is that besides associations between immuno-nutrition and CRC outcomes also the tumour micro-environment was investigated, revealing potential underlying mechanisms.

However, I have some major concerns though:

- The study-design is not completely clear from the abstract and also not from the methods. It is a retrospective study with cases and controls. Apparently patients were given either immuno-nutrition or not. Based on what? Clinical practice? A previous RCT?

- Controls were considered ineligible for the protocol. Why? If the controls are patients with a good nutritional status, healthy BMI no involuntary weight loss etc. and therefore not eligible for IN wouldn`t this influence the results? In that case you are comparing to very different populations. In the results section you mention that the two groups were not different regarding epidemiological, clinical and pathological characteristics, though in my opinion some differences for example age, tumour location, MMR status and comorbidities were observed. Why were they not eligible for the IN? Please add this information to the manuscript.  

- Statistical analyses were not described in the paper itself. Please add this (referring to another paper is not sufficient).

- The results section is rather limited. For example no tables with numbers for the post-operative outcomes (heading 3.2) and the same holds for the results described under heading 3.3. It would be interesting to also add tables with the specific numbers for each group for all outcomes. This provide more information compared to just p-values. For example, an anastomotic leakage is a very serious complication and the p-value for the differences between groups is 0.07, it would be interesting to have more information on this, how many events occurred in each group? The same about the tumour characteristics, given information is very limited (only mentioned whether it is statistically significantly different or not).

- How well can you compare the pre-surgical biopsy with the post-surgical sample? Can be sampled from a different part of the tumour, how could this have influenced your results?

- The discussion is poorly written and should be entirely rewritten. The structure is unclear and some conclusions are not linked to the results. For example line 257-258, maybe you showed this, but this is not part of the research question and not described earlier in the manuscript. In line 260-270 new results are described (which should not be done in the discussion). In addition a lot of background information is provided (which is not a problem on its own), however, this is not linked to findings of this study. No recommendations for further studies were done. Also strength and limitations should be better described.

Other remarks:

- Line 130: RNA is not introduced before. Or are you referring to nucleotide (FNA; typo?) line 80. Nucleotide is not a nutrient right?

Author Response

In this very interesting study, the association between immune-enhancing nutrition and several outcomes in colorectal cancer patients, including the immune response, post-operative complications and survival was investigated. This is a intriguing and important topic. Although the sample size was rather small (50 cases and 50 controls), the study provide important insights into the relationship between immuno-nutrition and CRC outcomes. An important strength of this study is that besides associations between immuno-nutrition and CRC outcomes also the tumour micro-environment was investigated, revealing potential underlying mechanisms.

Dear Reviewer thank you very much for your criticisms and suggestions that have improved the quality of the manuscript. We extensively re-write the following sections as you suggested: material and methods, results, discussion and conclusion. English language was revised by a native speaker.

However, I have some major concerns though:

- The study-design is not completely clear from the abstract and also not from the methods. It is a retrospective study with cases and controls. Apparently patients were given either immuno-nutrition or not. Based on what? Clinical practice? A previous RCT?- Controls were considered ineligible for the protocol. Why? If the controls are patients with a good nutritional status, healthy BMI no involuntary weight loss etc. and therefore not eligible for IN wouldn`t this influence the results? In that case you are comparing to very different populations. In the results section you mention that the two groups were not different regarding epidemiological, clinical and pathological characteristics, though in my opinion some differences for example age, tumour location, MMR status and comorbidities were observed. Why were they not eligible for the IN? Please add this information to the manuscript.  

We better clarified the methodology of the study design and addressed all your issues in the present form of the manusccript.

- Statistical analyses were not described in the paper itself. Please add this (referring to another paper is not sufficient).

We apologize but we have a limited number of characters for the papers, thus we summarized some sections. However, in the present form of the manuscript, we better described the statistcal analysis.

  • The results section is rather limited. For example no tables with numbers for the post-operative outcomes (heading 3.2) and the same holds for the results described under heading 3.3. It would be interesting to also add tables with the specific numbers for each group for all outcomes. This provide more information compared to just p-values. For example, an anastomotic leakage is a very serious complication and the p-value for the differences between groups is 0.07, it would be interesting to have more information on this, how many events occurred in each group? The same about the tumour characteristics, given information is very limited (only mentioned whether it is statistically significantly different or not).

In the present form of the manuscript we added three tables to show results regarding heading 3.2 and 3.3. In the previous form of the paper we did not show TME by tables as presenting results for each patient provide low quality tables as you can appreciate.

- How well can you compare the pre-surgical biopsy with the post-surgical sample? Can be sampled from a different part of the tumour, how could this have influenced your results?

In the material and methods section we describe the endoscopic procedure for sampling. We omitted it in the previous version for the limited number of characters.

- The discussion is poorly written and should be entirely rewritten. The structure is unclear and some conclusions are not linked to the results. For example line 257-258, maybe you showed this, but this is not part of the research question and not described earlier in the manuscript. In line 260-270 new results are described (which should not be done in the discussion). In addition a lot of background information is provided (which is not a problem on its own), however, this is not linked to findings of this study. No recommendations for further studies were done. Also strength and limitations should be better described.

We agree completely with the reviewer; accordingly, we extensively re-write the discussion. 

Other remarks:

  • Line 130: RNA is not introduced before. Or are you referring to nucleotide (FNA; typo?) line 80. Nucleotide is not a nutrient right?

We corrected it accordingly.

Round 2

Reviewer 2 Report

The authors addressed my previous comments sufficiently. The study design is clear now and the manuscript itself is well written.

Some minor remarks:

Line 114: should this read conventional nutrition instead of conventional nutritional?

Why was the coding for comorbidities and stage changed?

Quality of figure 4 is low (figures are a bit blurred)

Table 3 and 4 are not readable.

The immune-characteristics of the IN-group and the non-IN group at time of the biopsy is very different, while the biopsy is before the start of the immuno-nutrition treatment right? Can you explain (in the discussion) this difference? And how does this effect your results/conclusion? It is very clear form your results that IN modulates host immune responses to the tumor (at least clear differences in immune-cell activation/presence), however, I do not understand why the TME of the non-IN groups differs from the IN-group at baseline (biopsy). Or was IN already started at the time of the biopsy?

Can you elaborate a bit on future research needed?

Author Response

Reviewer 2 

The authors addressed my previous comments sufficiently. The study design is clear now and the manuscript itself is well written.

Dear Reviewer thank you for your precious time in reviewing our paper and providing valuable comments. It was your valuable and insightful comments that led to possible improvements in the current version.

Some minor remarks:

- Line 114: should this read conventional nutrition instead of conventional nutritional?

Thank you for pointing this out. This is a typo error which we accordingly corrected.

- Why was the coding for comorbidities and stage changed?

The changes have been made to simplify and improve the readability of the tables. Some groupings were too small for reliable statistical comparison. The results remained essentially unchanged. For comorbidities, less than or equal to two is generally considered acceptable for low-risk surgery.

- Quality of figure 4 is low (figures are a bit blurred)

Taking the reviewer's suggestion, we provide figures with better sharpness.

- Table 3 and 4 are not readable.

As we explain in the previous letter of response, we did not include both tables in the first version of the manuscript as they resul not readable. We know that the Reviewers needed the table to check our findings patient by patient. By know, we prefer to delete the tables from the final version if you agree.

We agree on the poor readability of tables 3 and 4 which were inserted at the request of the reviewer, who rightly wanted to be informed in detail about the evaluations carried out. We still think these tables are ultimately unnecessary and that the results will be better and immediately explained by the novel figures we introduce in the revised version of the manuscript. However, if it is still deemed necessary, tables 3 and 4 can be inserted again, providing them with explanatory captions.

 - The immune-characteristics of the IN-group and the non-IN group at time of the biopsy is very different, while the biopsy is before the start of the immuno-nutrition treatment right? Can you explain (in the discussion) this difference? And how does this effect your results/conclusion? It is very clear form your results that IN modulates host immune responses to the tumor (at least clear differences in immune-cell activation/presence), however, I do not understand why the TME of the non-IN groups differs from the IN-group at baseline (biopsy). Or was IN already started at the time of the biopsy?

IN started after the time of the biopsy. The cases enrolled in the two groups were not selected and were consecutive; therefore, we believe that the observed differences are mainly due to chance and possibly reflect the relatively small number of the patients evaluated. Therefore, we do not think that the differences appreciated in TME of both groups at baseline might affect our conclusions. On the other hand, the primary objective of the study was to evaluate the effects of immunonutrition on TME. For that reason, in the latest version of our article proposed it was considered appropriate to focus attention on this aspect, omitting graphical representations that are not relevant for the purposes of the study. To this aim we modified Figure 3 and Figure 4.

- Can you elaborate a bit on future research needed?

We believe that the results obtained by immunohistochemistry should be integrated with an analysis of the cytokines involved in the immune reaction, such as IL-10, IL-17, IL-6, IFNg, TNFa as well as with the evaluation of the same lymphoid populations by citofluorimetry in the patients’ blood. We also believe we should collect a higher number of cases with a more detailed survival analysis and possibly longer follow-up. We comment on this in the conclusion.